# Deep ocean warming-induced El Niño changes

Geon-Il Kim [1], Ji-Hoon Oh [1], Na-Yeon Shin [1], Soon-Il An [2], Sang-Wook Yeh [3], Jongsoo Shin [4] & Jong-Seong Kug [1,5] ✉

The deep ocean, a vast thermal reservoir, absorbs excess heat under greenhouse warming, which ultimately regulates the Earth's surface climate. Even if $CO_2$ emissions are successfully reduced, the stored heat will gradually be released, resulting in a particular pattern of ocean warming. Here, we show that deep ocean warming will lead to El Niño-like ocean warming and resultant increased precipitation in the tropical eastern Pacific with southward shift of the intertropical convergence zone. Consequently, the El Niño-Southern Oscillation shifts eastward, intensifying Eastern Pacific El Niño events. In particular, the deep ocean warming could increase convective extreme El Niño events by 40 to 80% relative to the current climate. Our findings suggest that anthropogenic greenhouse warming will have a prolonged impact on El Niño variability through delayed deep ocean warming, even if $CO_2$ stabilization is achieved.

Over the past few centuries, the Earth has been warming at an unprecedented rate due to human activities, raising serious concerns about the risks this poses[1]. In response to these potential dangers, the 2015 Paris Agreement enshrined the global goal of limiting the global temperature rise to below 2 °C, and ideally to 1.5 °C. Achieving the objective of limiting climate change requires balancing carbon dioxide emissions with equivalent carbon absorption[2] and potentially reducing $CO_2$ concentrations below current levels to avoid climate tipping[3]. However, even if we were to reduce $CO_2$ in the atmosphere to pre-industrial levels, the climate system would not return to its original state. In other words, the climate system would change to a different state with the same level of anthropogenic forcing, which is referred to as climate irreversibility. Therefore, understanding climate irreversibility is crucial for assessing the ability of the climate to recover from anthropogenic global warming and for designing climate policy[4].

Accordingly, including the Carbon Dioxide Removal Model Intercomparison Project (CDRMIP)[5], several idealized modeling experiments with different approaches have been conducted to explore the potential of mitigating $CO_2$ forcing. The irreversibility of several key climate components has been identified, such as surface temperature[6-8], sea level[9-11] precipitation[12-14], the Atlantic Meridional Overturning Circulation (AMOC)[15], the Antarctic ice sheet[16] and the Intertropical Convergence Zone (ITCZ)[17]. Concerning the mechanism of irreversible climate change, the role of the deep ocean has recently been highlighted[18]. The deep ocean absorbs and stores large amounts of thermal energy and facilitates the slow transport of heat from the ocean surface to its interior, acting as a heat reservoir[19]. The heat stored in the deep ocean will be released at a slower rate than it has been accumulated, leading to irreversible warming of the deep ocean over several centuries and regional irreversibility of sea surface temperature (SST) and precipitation[18]. This irreversible warming of the SST primarily occurs in the high latitudes and the equatorial eastern Pacific, where the oceans are weakly stratified to allow vigorous heat release through both vertical upwelling and mixing from the deep ocean to the surface. The efficiency of energy transfer from the deep ocean to the surface is closely related to ocean stratification, so deep ocean warming modulates irreversible SST changes in a unique pattern, e.g., El Niño-like warming.

The El Niño-Southern Oscillation (ENSO) is the most dominant interannual climate variability, affecting global weather and climate[20],

[1]School of Earth and Environmental Sciences, Seoul National University, Seoul, South Korea. [2]Department of Atmospheric Sciences, Yonsei University, Seoul, South Korea. [3]Department of Marine Sciences and Convergent Technology, Hanyang University, Ansan, South Korea. [4]Woods Hole Oceanographic Institution, Woods Hole, Massachusetts, United States of America. [5]Interdisciplinary Program in Artificial Intelligence, Seoul National University, Seoul, South Korea. ✉e-mail: jskug1@gmail.com

and understanding changes in ENSO in the future climate is important because of its large environmental, economic, and social impacts[21]. In recent decades, various studies have investigated ENSO changes under greenhouse warming in terms of its intensity, asymmetry, duration, spatial pattern, diversity, and teleconnections[22–25]. For example, the greenhouse warming has led to strong surface feedback, resulting in more Central Pacific (CP) El Niño[26,27], as well as an enhanced nonlinear atmospheric response in the Eastern Pacific (EP), leading to more extreme EP El Niño[26,28]. As such, the changes in ENSO variability have been studied extensively, while ENSO changes under $CO_2$ removal scenarios are not fully understood. For example, the SST variability over the eastern Pacific[29,30] and frequency of convective extreme El Niño[31] has increased due to the southward shift of the ITCZ, despite the return of $CO_2$ concentration to current levels. During the climate recovery process, heat released from the deep ocean leads to an effective warming of the equatorial eastern Pacific[32], implying a link between deep ocean warming and ENSO changes. However, this possibility has not yet been investigated.

Accordingly, the changes in ENSO during the restoring period after the return of $CO_2$ levels to their present-day climate conditions were investigated. During this period, the Earth's climate was incompletely recovered and warmer compared to the current climate. This study will investigate the role of irreversible deep ocean warming in ENSO changes. This can provide a perspective on ENSO changes attributed to deep ocean warming, whereas previous studies have interpreted ENSO changes mainly in terms of ocean-atmosphere interactions and upper ocean changes.

## Results

### The changes in tropical Pacific mean state

To simulate the climate after mitigating global warming by reducing $CO_2$ emissions, the Community Earth System Model (CESM) was run with constant atmospheric $CO_2$ (restoring) after it was increased for 140 years, then symmetrically decreased (see "Methods" and Supplementary Fig. 1). The mean state change was quantified by calculating the difference between the restoring period, defined as the entire 220 years of the restoring simulation, and the present-day (PD) period, defined as the last 400 years of the PD simulation. Figure 1a, b show the changes in the tropical mean state from the PD period. Even when the

$CO_2$ concentration returns to current levels, the tropical mean state shows considerable differences. SST anomalies (SSTA) show positive signals across the tropics, indicating incomplete recovery and a warmer state compared to the present climate. In particular, El Niño-like warming (i.e., stronger warming in the eastern Pacific) is evident in the spatial pattern of the SST difference, which is similar to the slow response to global warming[33,34].

The increase in precipitation in the tropics and the decrease in the subtropics indicate a southward shift of the ITCZ in the Pacific (Fig. 1b). The latitudinal position of the ITCZ is linked to the energy transport across the equator, which is regulated by the meridional energy exchange between the hemispheres[35]. As the climate recovers after $CO_2$ reaches current levels, the Northern Hemisphere (NH) cools much faster than the Southern Hemisphere (SH) (Supplementary Fig. 2). The larger heat capacity of the SH[17] and the pronounced heat release from the Southern Ocean[18] contribute to a relatively slow cooling rate in the SH. Therefore, the reduction in interhemispheric thermal contrast drives the shift of the ITCZ toward the relatively warmer SH. The southward shift of the Pacific ITCZ is accompanied by an increase in precipitation and a weakening of the trade winds over the equator, which further amplifies El Niño-like warming through the Bjerknes feedback. The mean state change shown in Fig. 1a, b will be eventually determined as a result of this positive feedback.

Previous study[18] suggested that the SST response in the restoring period originates from deep ocean warming and climatological ocean stratification. To investigate the influence of deep ocean warming on changes in the tropical mean state, three initial warming experiments (IW_EXPs) were conducted. Specifically, the IW_EXPs were conducted with a constant atmospheric $CO_2$ level but with horizontally uniform vertical profiles of ocean temperature and salinity anomalies added to the ocean model initial conditions. These IW_EXPs allow us to explore the impacts of deep ocean warming on the mean state of the tropical ocean and subsequent alterations in ENSO characteristics (see "Methods" for a detailed description of the IW_EXPs).

The mean state change was quantified by calculating the difference between the average for the last 100 years after 50 years from the IW_EXPs and the PD period. Figure 1c, d shows the SST and precipitation responses when the initial ocean warming is added below 700 m (IW_be700). Although spatially uniform warming is only added below

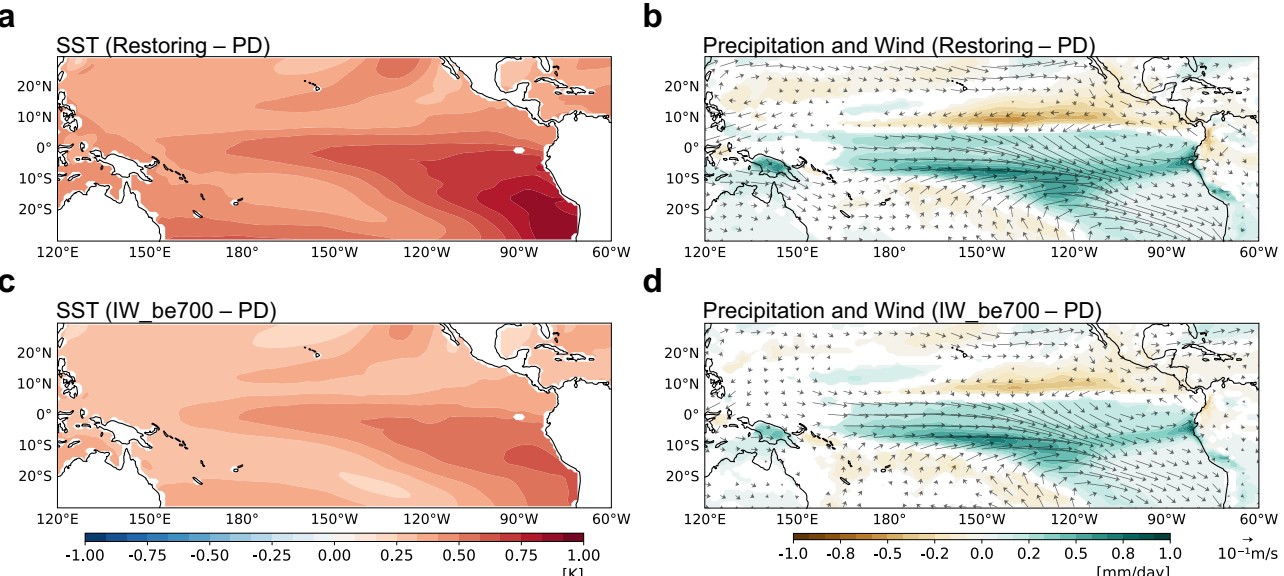

**Fig. 1 | The changes in the tropical mean state.** The difference in (**a**) sea surface temperature (SST), (**b**) precipitation, and wind at 850hPa between the restoring and the present day (PD) period. The difference in (**c**) SST, (**d**) precipitation, and wind at 850hPa between the initial warming added below 700 m experiment (IW_be700) and the PD period. Only significant values at the 95% confidence level using the bootstrap test are shown.

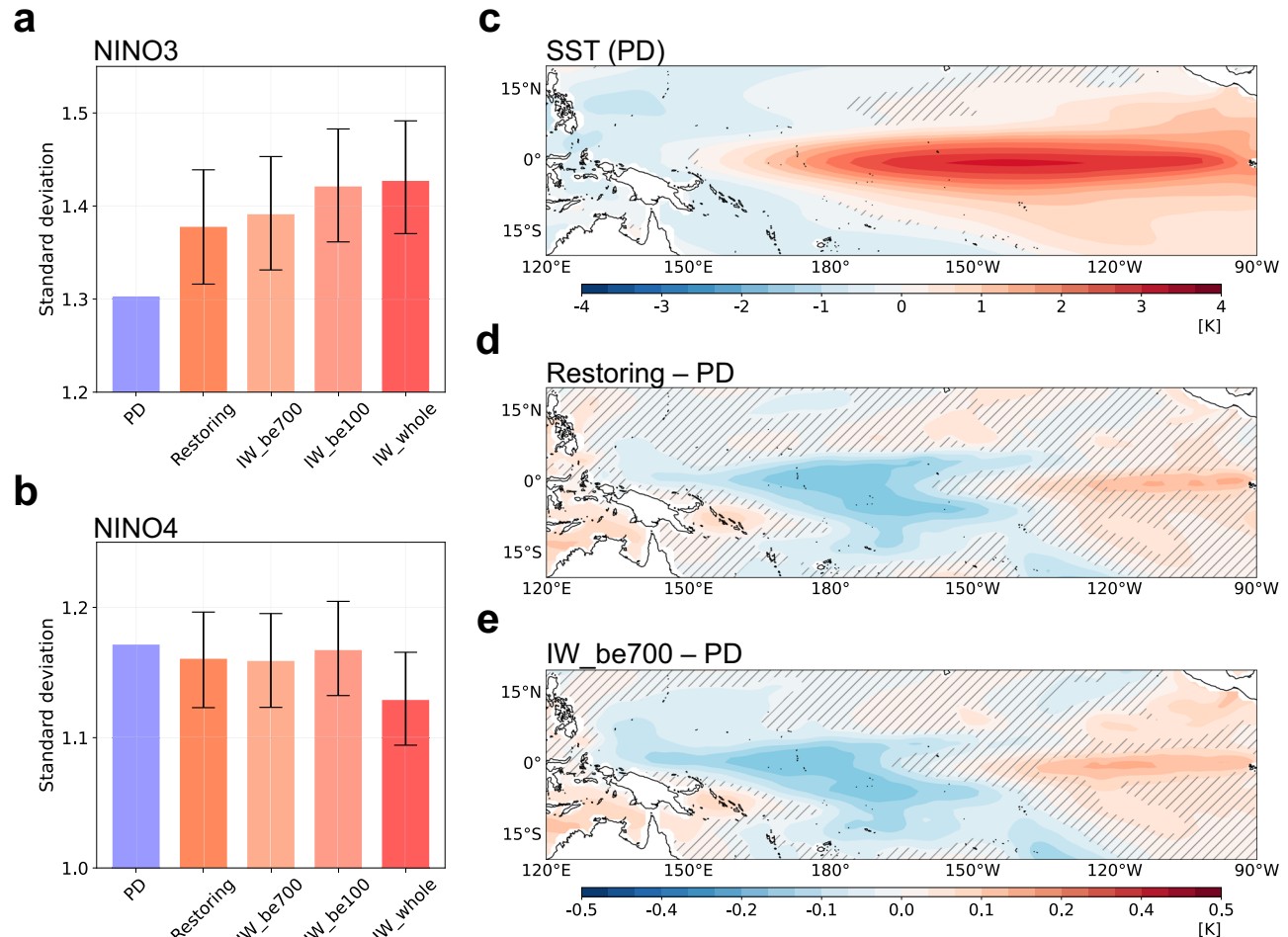

**Fig. 2 | The changes in El Niño-Southern Oscillation amplitude.** The standard deviation of the December-January-February (DJF) (**a**) Niño3 and (**b**) Niño4 sea surface temperature anomaly (SSTA). **c** The spatial pattern of El Niño SSTA during DJF for the present day (PD) period. El Niño was defined using the 1 standard deviation (STD) threshold of the DJF Niño3 SSTA. As in (**c**) but for the difference from the PD period to (**d**) restoring and (**e**) initial warming added below 700 m experiment (IW_be700). Error bars represent the 95% confidence interval using the bootstrap test. Stippled areas are regions that are not significant at the 95% confidence level of the Student's *t* test in the PD period and for the differences of restoring and IW_be700 using the bootstrap test.

700 m, the initial deep ocean warming leads to a particular SST pattern. Interestingly, the pattern is almost identical to that in the restoring period, with a high pattern correlation of 0.95 over the tropical Pacific. Similarly, the pattern correlation for the precipitation pattern is 0.97. Furthermore, the other IW_EXPs, adding initial ocean warming in the whole ocean depth (IW_whole) and below 100 m (IW_be100), also show similar changes in SST and precipitation, although their magnitudes are different (Supplementary Fig. 3), supporting strong robustness. These results suggest that deep ocean warming triggers specific patterns in the tropical mean state, such as El Niño-like warming and southward shift of the ITCZ. As reported in the previous study, the deep ocean acts as a stove, continuously supplying heat to the ocean surface, and the efficiency of this process depends on ocean stratification[18]. As a result, the energy from the deep ocean is more effectively transferred to the surface in the eastern Pacific, where stratification is weaker and climatological upwelling prevails[18]. The initial warming over the eastern Pacific (Supplementary Fig. 4) will be intensified by the Bjerknes feedback and the southward shift of the Pacific ITCZ.

### The changes in the characteristics of the ENSO

The El Niño-like warming and the southward shift of the ITCZ have the potential to intensify the variability of the ENSO in the eastern Pacific[28]. Therefore, we hypothesized that deep ocean warming may contribute to changes in ENSO characteristics by the tropical mean state changes. To examine this possibility, we first checked the amplitudes of the Niño indices (Fig. 2a, b, see "Methods"). The standard deviation (STD) of the Niño3 SSTA increases significantly, while the Niño4 SSTA remains unchanged except for IW_whole. Interestingly, the STDs of the Niño3 SSTA in IW_be700, the IW_be100, and IW_whole experiments consistently increase relative to the PD period, similar to the change in the restoring period. The largest increases are in the order of: IW_whole, IW_be100, and IW_be700, which are proportional to the tropical mean state warming (Supplementary Fig. 3). These results suggest that deep ocean warming enhances the SST variability over the eastern Pacific.

Figure 2c shows the El Niño composites for December-January-February (DJF) during the PD period. Here, El Niño events were defined based on 1 STD of the Niño3 SSTA (see "Methods"). The SSTA pattern shows positive anomalies in the equatorial Pacific and the Niño3 SSTA during the PD period, with a maximum near 141°W. During the restoring period, the Niño3 SSTA increased with the maximum near 134°W, indicating an increased ENSO amplitude and an eastward shift (Supplementary Fig. 5). The SSTA difference from the PD period shows a zonal dipole pattern (Fig. 2d), showing that the equatorial eastern and western Pacific regions have opposite signs, consistent with the increase in STD of the Niño3 (Fig. 2a). In addition, Coupled Model Intercomparison Project Phase 6 (CMIP6) models further support our

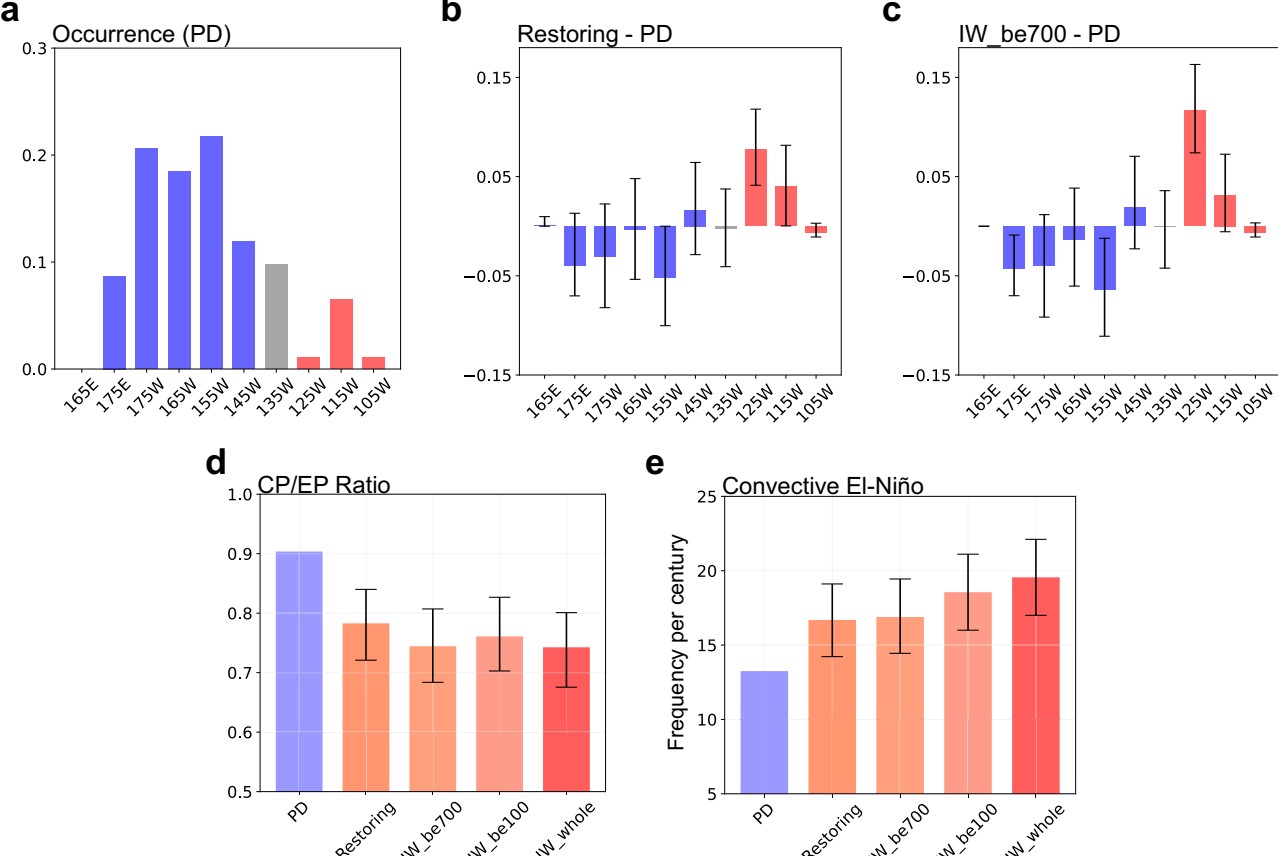

**Fig. 3 | The changes in El Niño-Southern Oscillation's flavor. a** Histogram (normalized occurrences) of El Niño sea surface temperature anomaly (SSTA) centers during December-January-February (DJF) for the present day (PD) period. As in (**a**) but for the difference from the PD period to (**b**) restoring and (**c**) initial warming added below 700 m experiment (IW_be700). **d** Ratios of frequencies of

Central Pacific (CP) El Niño events proportion to the total number of CP and Eastern Pacific (EP) El Niño events. **e** The frequency of convective El Niño, which is defined as absolute precipitation exceeding a threshold of 5 mm/day during the DJF season. Error bars represent the 95% confidence interval using the bootstrap test.

argument (see Supplementary Note and Supplementary Fig. 6). Interestingly, IW_be700 simulates quite a similar difference in SSTA from the PD period (Fig. 2e). Although only uniform deep ocean warming is added to the initial condition, the model simulates mostly identical El Niño SSTA pattern to that in the restoring period, suggesting that the warming of the deep ocean is the most critical factor in changing El Niño characteristics in the restoring period.

The eastward shift of the maximum in the ENSO synthesis suggests a systematic change in ENSO diversity. To investigate this, we analyzed changes in the spatial diversity of El Niño using a method that detects the longitude of the SSTA peak during DJF (see "Methods"). The zonal distribution of SSTA peaks shows a bimodal structure with maxima at 155 W and 115 W (Fig. 3a), although the model tends to simulate more CP El Niño compared to the observational distribution[26] (see Fig. 1a, b). Based on this result, we can divide El Niño events into two distinct types: CP El Niño, with maximum SSTA between 165°E − 145°W, and EP El Niño, between 125°W − 105°W (Supplementary Fig. 7). In addition, to make a quantitative comparison, the CP ratio was defined as the CP events relative to the sum of CP and EP events. The change in SSTA peak occurrence shows that EP El Niño events are more frequent in the restoring period (Fig. 3b), consistent with the decrease in CP ratios in the restoring period relative to the PD period (Fig. 3d). The decrease in the frequency of CP El Niño events is consistently simulated in the IW_be700 (Fig. 3c), IW_be100, and IW_whole experiments (Supplementary Fig. 8). These results suggest that heat release from the deep ocean prefers to more frequent EP El Niño. In addition, the distribution for peaks of La Niña events also shows quite similar

changes to that of El Niño events in the restoring period and all IW_EXPs (Supplementary Fig. 9).

EP El Niño is mostly stronger than CP El Niños[36,37] and tends to be associated with extreme events, which are directly linked to anomalous climate conditions worldwide[31,38]. To assess the risk by changes in the extreme case of El Niño, we have adopted the concept of a convective extreme El Niño (see "Methods"), which will be more closely related to anomalous atmospheric circulation[28,31]. The changes in the frequency of the convective El Niño from the PD period are shown in Fig. 3e. During the PD period, convective El Niño occurs about 13 times per 100 years, accounting for 10% of all El Niño events. On the other hand, the frequency of convective El Niño increased by 40 to 80% in warmer climates. In addition, the CMIP6 models provide additional support for our argument (see Supplementary Note and Supplementary Fig. 6).

### The changes in ENSO feedback

So far, we have shown that the deep ocean warming induces the tropical mean state change, resulting in more EP El Niño and convective El Niño. It is important to understand what physical processes lead to these changes in the ENSO characteristics. During the restoring period, there is a significant increase in the equatorial ocean temperature below the 300-m depth (Supplementary Fig. 10a). This result is consistent with all IW_EXPs (Supplementary Fig. 10b–d), suggesting a release of heat from the deep ocean to the surface. In the eastern Pacific, a region of weak ocean stratification and intense upwelling, heat is much more efficiently transferred to the surface. The surface warming is further amplified by the Bjerknes feedback so that the surface warming is greater than the

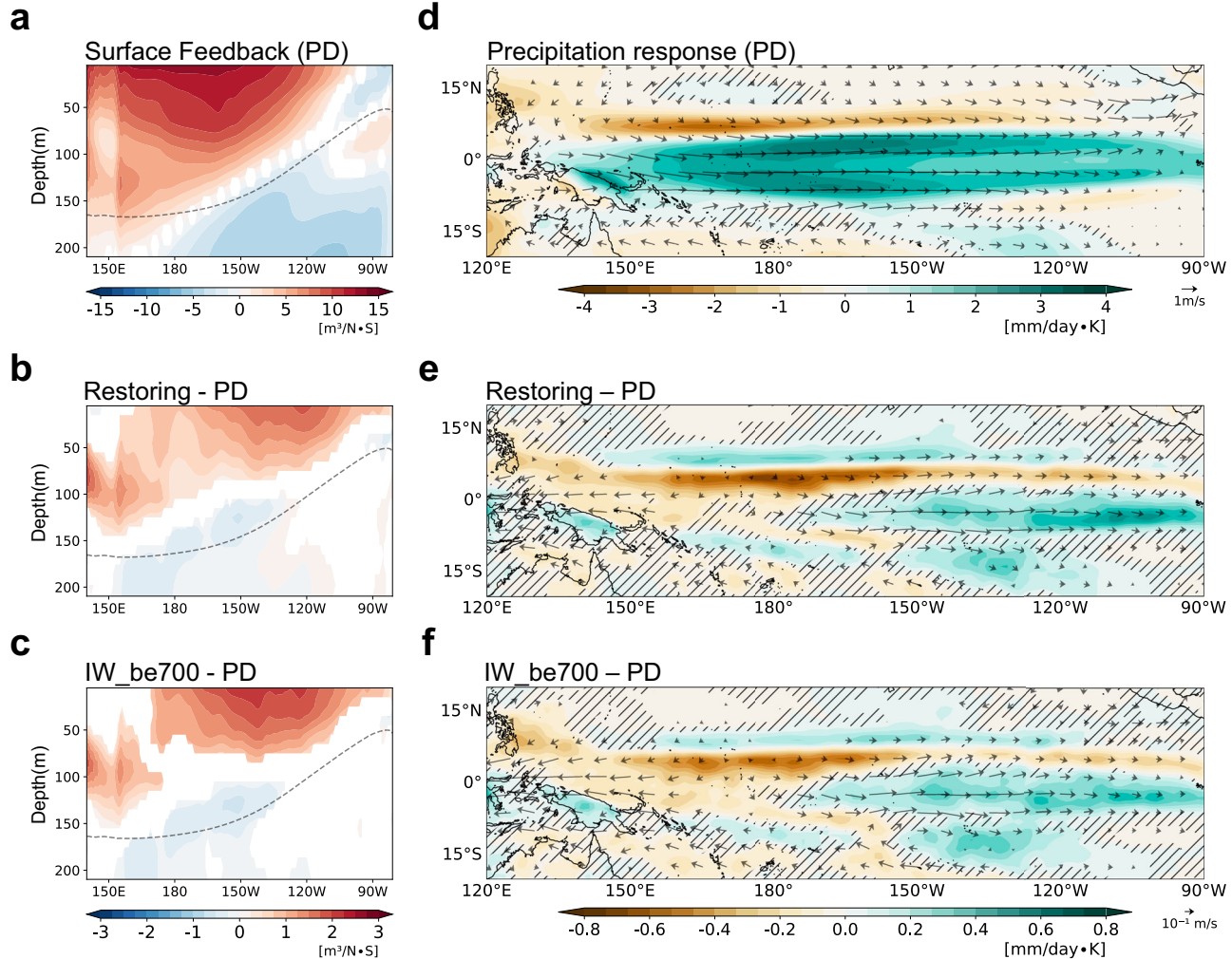

**Fig. 4 | The changes in El Niño-Southern Oscillation feedback. a** The regression coefficients of zonal current onto averaged wind stress over the equatorial Pacific (120°E–90°W, 5°S–5°N) during December-January-February (DJF) for the present day (PD) period. As in (**a**) but for the difference from the PD period to (**b**) restoring and (**c**) initial warming added below 700 m experiment (IW_be700). The dashed line represents the climatological thermocline depth. Only significant values at the 95% confidence level using the bootstrap test are shown. **d** The regression coefficients of precipitation (shading) and 850hPa wind (vector) onto Niño3 sea surface temperature anomaly (SSTA) during DJF for the PD period. As in (**d**), but for the difference from the PD period to (**e**) restoring and (**f**) IW_be700. Stippled areas are regions that are not significant at the 95% confidence level of the Student's *t* test in the PD period and for the differences of restoring and IW_be700 using the bootstrap test.

subsurface warming though the warming originated from the deep ocean warming. Therefore, the difference in the vertical temperature gradient is positive in the upper layer, suggesting enhanced ocean stratification (Supplementary Fig. 10e–h).

Enhanced stratification leads to strong air-sea coupling, indicating that the surface-layer responses to a given wind anomaly are amplified. To show enhanced surface-layer responses, we performed a linear regression analysis of the zonal current response to the zonal wind stress anomaly (120°E-90°W) during DJF (Fig. 4a). During the PD period, the current response to wind stress shows a stronger eastward flow in the surface layer, which gradually weakens with depth. Below the thermocline, there is a westward current response, indicating a vertically baroclinic structure. In the restoring period, the eastward current response in the surface layer is intensified (Fig. 4b), indicating stronger zonal advective feedback. Note that the stronger zonal current response is particularly distinctive in the eastern Pacific. In addition, the equatorial upwelling and thermocline responses are also intensified in the eastern Pacific (Supplementary Fig. 11). This enhanced surface-layer feedback plays a critical role in the enhanced eastern Pacific SST variability. The IW_be700 also simulates enhanced

feedback in the surface layer (Fig. 4c). This result suggests that deep ocean warming leads to stronger oceanic responses in the eastern Pacific upper layer. The other two IW_EXPs show similar changes in the zonal current response (Supplementary Fig. 12a, b), supporting strong robustness.

The ocean stratification is responsible for the intensified surface-layer feedback, but it cannot account for the eastward shift of the responses. Previous studies suggested that the El Niño-like mean conditions not only thermodynamically enhance the response of precipitation to equatorial SST but also provide favorable conditions for the shift of anomalous atmospheric heating due to increased mean trade wind convergence[39]. For example, when the eastern Pacific is in a more moist state, the response of wind and precipitation to SSTA shifts eastward[40]. The eastward shift of the wind response to ENSO anomalies strengthens the thermocline and zonal advection feedback, ultimately increasing the eastern Pacific SST variability[41,42]. As mentioned earlier, the El Niño-like warming induces an eastward shift of the ascending branch of the Walker circulation and an equatorward shift of the ITCZ (Fig. 1), which strengthens the shift in the surface-layer current response.

To analyze the eastward shift of the ENSO feedback, we calculated the regressed precipitation and wind onto the Niño3 SSTA during DJF (Fig. 4d). As expected, the precipitation pattern shows positive anomalies associated with westerlies in the equatorial Pacific. The changes in the precipitation and wind response from the PD period indicate an eastward extension of the response, with positive anomalies observed in the eastern Pacific and negative anomalies in the western Pacific (Fig. 4e). The eastward shift of the wind response to the SSTA contributes to the enhanced Ekman feedback. In addition, the strong equatorial wind stress response enhances the thermocline feedback by intensifying the equatorial Kelvin wave responses. In addition, increases in the meridional slope of the wind stress (Fig. 4e), resulting in a stronger off-equatorial Rossby wave response. The IW_EXPs consistently simulate the precipitation response eastward shifted compared to the PD period (Fig. 4f and Supplementary Fig. 12c, d). The pattern of precipitation change is almost identical to that of the restoring period, with a pattern correlation of 0.95. These results suggest that the warming of the deep ocean produces the ENSO system eastward in the restoring period. Note that the westerly anomaly is stronger at 150-120°W, consistent with where the surface current response increases (Fig. 4b, c). In short, the occurrence of more EP El Niño can be attributed to the eastward shift of the ENSO feedback system caused by El Niño-like warming.

## Discussion

In this study, we showed that deep ocean warming induces changes in ENSO characteristics by altering the tropical mean state change. If net-zero carbon emissions are achieved, heat release from the deep ocean to the surface layer can cause El Niño-like warming[18], leading to an increase in EP-type ENSO and extreme El Niño events. These changes are driven by ocean stratification and El Niño-like warming, which enhance zonal advective, Ekman, and thermocline feedbacks in the eastern Pacific, despite increased thermal damping (Supplementary Fig. 13). Our findings consistently show that deep ocean warming is the primary driver of these ENSO changes, as observed in the IW_EXPs. Most notably, our research indicates that even if $CO_2$ levels are reduced to current levels, the ongoing release of energy from the deep ocean into the surface will likely lead to more frequent and extreme El Niño events, posing increased risks to humanity.

The hysteresis of the precipitation and SST remains over the tropical Pacific even with a rapid $CO_2$ reduction[8], indicating that changes in ENSO are inevitable. The strength of the irreversibility depends on the rate of $CO_2$ reduction, which contributes to the amount of energy stored in the ocean. The slower the rate of reduction, the more heat will be stored in the ocean and the greater the warming of SST when $CO_2$ returns to current levels[43]. The results from the IW_EXPs show that the stronger the El Niño-like warming, the greater the increase in SST variability over the eastern Pacific (Fig. 2a). These results indicate that a slower rate of decline will result in a more prolonged and pronounced warming of the tropical Pacific SST, which may potentially lead to a more intense El Niño event.

The mechanism underlying ENSO changes induced by deep ocean warming differs significantly from that associated with anthropogenic greenhouse warming (Supplementary Fig. 14). While global warming elevates ocean surface temperatures, leading to heightened ocean stratification, especially in the central Pacific, it consequently intensifies the ocean current response in this region and promotes more frequent CP-type El Niño events. Conversely, deep ocean warming begins from a state of initially reduced stratification, limiting its stratification effect. Moreover, El Niño-like warming is particularly pronounced due to robust vertical mixing in the eastern Pacific and a southward shift of the ITCZ. This leads to an eastward shift in the ENSO system and reinforces surface-current response in the eastern Pacific, culminating in a higher frequency of EP-type El Niño events.

Numerous coupled models have projected an El Niño-like pattern as a response to global warming. However, observed warming patterns in recent decades have shown a tendency towards La Niña-like conditions rather than El Niño-like[44]. This discrepancy between model predictions and actual observations may arise from several factors, including cold-tongue biases[45], an underestimation of the inter-basin warming contrast[46], and an over-weakened Walker circulation[47], but the exact reasons remain uncertain. Our study suggests that deep ocean warming could be an additional factor to consider. If models overestimate the accumulation of heat in the deep ocean compared to actual observations, this could lead to biases in SST projections. Given the profound influence of changes in the tropical mean state on ENSO, it's essential to reassess future ENSO changes in light of anticipated El Niño-like warming trends.

## Methods

### Definition of ENSO

The Niño3 and Niño4 SSTA, defined as the area-averaged SSTA in the Niño3 region (5°S-5°N, 150-90°W) and Niño4 region (5°S-5°N, 160°E-150°W) are used to describe the amplitude of ENSO, respectively. The SSTA was calculated by removing the linear trend and the monthly climatological mean for each experiment and ensemble. We utilize a threshold of $\pm 1$ STD of the DJF mean Niño3 SSTA during the PD period to define ENSO events. To define convective extreme El Niño, we employed the method used in previous studies[31,48–50], wherein convective extreme El Niño was identified as absolute precipitation exceeding a threshold of 5 mm/day during the DJF season.

### Detecting center method

To identify the center of the ENSO, we adopt the method used in the previous study[26], focusing specifically on the detection of its longitudinal peak: first, we calculate a moving average of the DJF SSTA over a 30° longitude window, moving at 10° intervals from 150°E to 90°W (10 points in total). We then identify the points where the moving average DJF SSTA is greater than 1 STD or less than $-1$ STD. The STD is calculated on the basis of DJF SSTA values above 150°E-90°W during the PD period. Among the selected points, the point with the largest absolute value is taken as the peak of El Niño (or La Niña). The longitude where the peak is located is taken as the center of ENSO.

### Model

This study used the CESM, Version 1.2, which comprises several components, including the Community Atmospheric Models Version 5 (CAM5), the Community Land Model Version 4 (CLM4), the Community Ice Code Version 4 (CICE4), and the Parallel Ocean Version 2 (POP2). The CAM5 and CLIM4 components were characterized by a horizontal resolution of about 1° and 30 vertical levels, while the CICE4 and POP2 components used a nominal horizontal resolution of 1° (with the 350 meridional resolutions being about 1/3° near the equator) and 60 vertical ocean levels.

### Design of the ramp-up and ramp-down experiment

The present and future climate simulations were carried out. The first simulation was a present-day scenario with a constant atmospheric $CO_2$ concentration (at the current level of 367 ppm) for 900 years (Supplementary Fig. 1). The second simulation was a $CO_2$ ramp-up and ramp-down scenario that started with different initial conditions from the present-day simulation (28 in total). This experimental design was similar to the 1pctCO2-cdr scenario used in the CDRMIP[4], except for the initial $CO_2$ level, which was set to the pre-industrial level (284.7 ppm). In these experiments, we gradually increased the atmospheric $CO_2$ concentration at a rate of 1% per year until it reached four times the initial value (1469 ppm) over a period of 140 years. We then reduced the $CO_2$ concentration symmetrically at the same rate for 140 years until it returned to the initial value (367 ppm). We then conducted a

recovery experiment with a constant $CO_2$ concentration (367 ppm) for 220 years to simulate a period of net-zero emissions. The second experiment consists of 28 ensemble members with different initial conditions for a total of 500 years in each ensemble.

## Design of the initial warming experiment

To investigate how deep ocean warming contributes to the change in ENSO characteristics, three types of initial warming experiments (denoted as IW_whole, IW_be100, and IW_be700) were conducted. These experiments were branched from the initial condition (year 2000) of each ensemble member and then integrated with the constant atmospheric $CO_2$ concentration of 367 ppm. In addition, horizontally uniform vertical profiles of the ocean temperature and salinity anomalies were added to the initial condition of the ocean model (year 2000). In other words, the initial temperature and salinity perturbations of IW_EXPs are horizontally identical in all ocean grids. The added initial anomalies are the global mean ocean temperature and salinity in the year 2280, when the atmospheric $CO_2$ level first returns to its original level (367 ppm). Note that, in addition to the temperature profile, a salinity anomaly profile was added to prevent any potential imbalance for running the model. To isolate the role of deeper ocean warming, the initial perturbation was added to the entire depth, below 100 m and below 700 m in IW_whole, IW_be100, and IW_be700, respectively. All experiments were integrated for 150 years with nine ensemble members. Note that these three IW_EXPs were also used in a previous study[18] (see Fig. 3).

## Data availability

The processed data are available at https://doi.org/10.6084/m9.figshare.25857760[51]. The CMIP6 data used in this study are available at https://esgf-node.llnl.gov/projects/cmip6. Source data are available at figshare.

## Code availability

The codes used in this study are available at https://doi.org/10.6084/m9.figshare.25857760[51].

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

## Acknowledgements

This work was supported by the Institute of Information & communications Technology Planning & Evaluation (IITP) grant funded by the Korea government (MSIT) [NO.RS-2021-II211343, Artificial Intelligence Graduate School Program (Seoul National University)]. J.-S. Kug was supported by the National Research Foundation of Korea (NRF) grant funded by the Korean government (NRF-2022R1A3B1077622). S.-I. An was supported by the National Research Foundation of Korea (NRF-2018R1A5A1024958).

## Author contributions

G.-I. Kim performed analyses, prepared the figures, and wrote the manuscript. J.-H. Oh conducted model experiments, and J.-S. Kug designed the research and wrote the manuscript. G.-I. Kim, J.-H. Oh, N.-Y. Shin, S.-I. An, S.-W. Yeh, J. Shin, and J.-S. Kug discussed the results and reviewed the manuscript.

## Competing interests

The authors declare no competing interests.
