## [Peer Review File · Nature Communications]

Deep Ocean warming-induced El Niño changesREVIEWER COMMENTS

Reviewer #1 (Remarks to the Author):

The authors used CESM model to perform idealized CO₂ ramp up and ramp down simulations in which atmospheric CO₂ increases at a rate of 1% per year to 4xCO₂, and then decreases at the same rate to 1xCO₂ (present-day level). The authors focused on the response of El Niño and the analyzed the role of deep ocean warming. By comparing El Niño events with present-day, the authors found that during the restoring period when atmospheric CO₂ returns to present-day level, Eastern Pacific El Nino events are intensified, precipitation in the tropical eastern Pacific increases, and convective extreme El Nino events also increase. By designing deep ocean warming simulations, the authors showed the importance of deep ocean warming in regulating El Nino response. This paper is well written. The simulations are well designed and the analysis is well done. I recommend its publication with minor modification.

How much heat is stored in the deep ocean and the resulting effect on El Nino response would depend on the rate of atmospheric CO₂ decrease. The author only performed ramp down simulations with a CO₂ decreasing rate of 1% per year. The implication of different CO₂ ramp down rate on the presented results should be discussed.

Specific comments:

Line 49: What results does 'here' refer to?

Lines 97-98: Please elaborate a bit more here of how the shift of the Pacific ITCZ partly contributes to the El Niño-like warming.

Lines 183-184: It might also be useful to show the vertical density gradient.

Lines 234-235: 'These changes are driven by ocean stratification and El Niño-like warming, ' It appears twice.

Line 288: How long does the current-day simulation last?

Line 309: How is the salinity anomaly profile determined?

Lines 379-380: Deep ocean warming induced irreversible climate changes Nature Climate Change, Nature Climate Change
For some reason, I could not find this article.

Fig. 1 caption (b) and (c) should exchange with each other. Or, the numbering of each panel should be changed.

Reviewer #2 (Remarks to the Author):

Kim et al. investigated the variations in ENSO during the recovery period after intensive global warming. Generally, this study showed some interesting results helping us to deepening understanding of ENSO, based on a series of numerical experiments. But I still have some concerns, which are listed below.

- 1) The mitigation scenario in the title is somewhat unrelated to this study.
- 2) "As the climate recovers after CO₂ reaches current climate levels, the Northern Hemisphere (NH) cools much faster than the Southern Hemisphere (SH)" The authors should give evidence to support this. What causes the different cooling rates? And what is the relative contribution of the NH-SH contrast and the El Nino-like warming pattern to the migration of the ITCZ?
- 3) "The initial warming over the eastern Pacific "in line 121. The authors may show the initial warming pattern in supplementary material, which is very helpful to understanding the formation processes of the El Nino-like warming pattern.
- 4) The authors stressed the intensification of the zonal-advection feedback. However, in previous studies, this feedback favors the CP El Ninos?
- 5) "The second experiment lasted 500 years and involved 28 ensemble members." Do you mean 500*28 years of integrations were conducted?
- 6) "A horizontally uniform temperature anomaly profile was added to the initial ocean state". Why uniform temperature anomalies are specified? This should be very different from those in recovery experiments.
- 7) Various feedbacks were diagnosed, but the authors did not mention which seasons they analyzed.

Reviewer #3 (Remarks to the Author):

March 2024,

A review of the manuscript NCOMMS-24-06651 "Deep Ocean warming-induced El Niño changes under climate mitigation scenarios" submitted by J.-S. Kug.

This study explored ENSO changes due to deep-ocean warming based on a CO₂ ramp-up and ramp-down experiment and present-day experiments initialized with horizontally uniform deep-ocean heat anomalies using CESM1.2. The authors examined not only the ENSO amplitude changes but also the diversity and feedback changes.

I have read several previous papers related to the CESM1.2 CO₂ ramp-up/down experiments. This paper aims to clarify the impact of the deep-ocean heat on the hysteresis of ENSO, which is important to the ENSO community. Although the results are interesting, I have some potentially critical concerns in their analyzing method. Because the present manuscript does not sufficiently

describe their method, I could not fully understand how the data was analyzed. I would like to ask the authors to revise the methodology and clarify my major concerns below.

Major comments:

1. The IW experiments are not at the equilibrium states, so the anomalous fields have to be carefully defined to compare with the PD experiment.

The last 100 years after 50 years are analyzed in the IW experiments and the PD period (Lines 106-107). However, it is clear in Fig. 3b of Oh et al. (2024), that the SST change is not steady at all. Thus, all the anomalous fields have to be carefully defined.

As the mean state remains changing for 150 years, it is important to describe it and consider removing its potential impact on the anomalous fields analyzed in this study. First of all, the definition of anomalies needs to be clarified. Then, how to remove the potential background changes from the anomalies needs to be described. For example, the Nino3 SSTA st.d. is easy to be affected by spurious slow anomalies due to background change. The SSTA pattern in Fig. 2 is also affected by the definition. Other main and supplementary figures are also sensitive to the definition of the anomalies.

Ideally, an initial condition large ensemble is required to define the anomalies from the changing background, but the experiments have only 9 ensembles for each of the IW experiments. Therefore, the method to define the anomaly is important.

2. The IW experiments are not fully described (Lines 104-105 and the Methods section), so it is not possible to reproduce the simulations based on the manuscript.

The method could be explained more in detail. At least, it must be clarified if the experiments are identical to the previous work (ref. 17).

And, what are the values of the uniform temperature anomaly profile and the corresponding salinity anomaly profile? How did you determine the salinity anomaly profile to avoid any potential density imbalance? Is there an initial condition's vertical gap at the 700-m (100-m) depth in IW_be700 (IW_be100)? The total heat content from the bottom to the surface may depend on the bottom topography, so probably the total heat content is not horizontally uniform --Does this affect the results? Because these points were unclearly written, I could not follow the detailed parts of the results.

3. Adding a schematic chart to explain the present conclusion may help readers understand what is different between the mechanism underlying ENSO changes induced by deep ocean warming and that associated with anthropogenic greenhouse warming (Lines 243-252).

4. The criteria for defining ENSO are not clear to me.

In Line 135, the authors said that "El Nino events were defined based on 1 st.d. of the Nino3 SST." Are these criteria the same among the experiments? Or, did you use different thresholds in each experiment? This difference may affect the patterns of the SSTA in Fig. 2c,d,e., for example.

5. The convective El Niño is not clearly defined in the text.

In Line 166, the authors described "the convective El Nino was defined as when Nino3 rainfall during the DJF season exceeds a threshold of 5 mm/day". Do the authors mean "Nino3 precipitation anomaly" instead of "the absolute precipitation"? If the absolute precipitation in the Nino3 region is used for defining the convective El Nino, the increase of the convective El Nino event occurrences is attributed to the mean state change to a certain extent.

6. It is unclear if there is inter-model uncertainty in the ENSO changes during the restoring period. In the Introduction, the authors imply that there are multi-model datasets of CDRMIP, which can be used to investigate the inter-model uncertainty of the present key results. I would suggest the authors conduct a multi-model analysis to indicate whether the major findings are independent of the models or dependent on the models.

Minor comments:

Title: I would say that this study analyzed El Nino changes under a single climate mitigation scenario in a climate model, so saying "under climate mitigation scenarios" does not seem representative of the study.

Lines 24-25: This sentence is too subjective and ambiguous to describe the implications of this study.

Line 51: The meaning of "residual upwelling" is unclear.

Line 64: I am quite unsure of the reason for citing ref. 28 (Ohba and Ueda 2009, JCLI <https://doi.org/10.1175/2008JCLI2334.1>).

Line 79: What is "the model"? The details can be found in the Methods section, but the model name could be provided in the main text for clarity.

Line 129: Actually, the Nino4 SSTA st.d. is decreased in IW_whole. How did you interpret this?

Line 130: The Nino3 SSTA st.d. is increased more in IW_be100 and IW_whole than in IW_be700 and Restoring. Why this happened could be explained in the text.

Line 136-137: How to calculate the central longitude is unclear. The Method section needs to be cited here.

Line 137: Any figure does not describe this text. A supplementary figure needs to be provided.

Lines 150-151: The observational distribution is not shown in any figure.

Figure 4: Is the figure calculated based on DJF? Please clarify the season used for this figure.

Line 190: Where is the thermocline in Fig. 4a-c?

Line 213: Define the Nino3 index. Do you mean Nino3 SSTA? In this manuscript, the index names are confusingly mixed (e.g., Nino3, Nino3 SST, Nino3 SSTA, Nino3 index).

Line 220: The results from the IW_be700 experiment are not mentioned yet. Figure 4f should appear in Lines 221-222.

Figure 2a,b: What is the season for the nino3 and nino4 SSTA standard deviations? Write in the caption.

Typos etc.:

Line 21: Use "Intertropical Convergence Zone" or "intertropical convergence zone" instead of "Intertropical convergence zone".

Line 84: Do you mean "The present-day period" instead of "the restoring period"?

Line 129, etc.: Use "Niño3 SSTA" instead of "Niño3 SST"? Check the manuscript overall.

Line 130, 141, 496, etc.: Use "Niño3 SSTA" instead of "Niño3"? Check the manuscript overall.

Line 188 etc.: Use "wind stress" instead of "windstress" overall.

Line 210: Replace "Fig" with "Fig".

Lines 235-236: This sentence repeats the previous sentence in Lines 234-235.

Line 272: "STD" is not defined.

Lines 331, 361: Check the titles of these references. "CO2" instead of "CO 2". And, "2°C" instead of "2 C".

References (e.g., ref. 14): The authors' names are not consistent in the references. for example, JS, or J-S? Check all references for consistency.

Line 17: Replace "cPhaden" with "McPhaden".

Line 380: Add a period at the end of the title.

Reference 17: I could not find ref. 17. Has it changed from the cited one to the following published paper? Oh, JH., Kug, JS., An, SI. et al. Emergent climate change patterns originating from deep ocean warming in climate mitigation scenarios. *Nat. Clim. Chang.* (2024).

<https://doi.org/10.1038/s41558-024-01928-0>

Reference 31: I could not reach this reference (Pathirana et al. 2022). Is the reference changed from the cited one to the following paper? Gayan Pathirana et al. Increase in convective extreme El Niño events in a CO2 removal scenario. *Sci. Adv.* 9, eadh2412 (2023). DOI:10.1126/sciadv.adh2412

Response to Reviewer #1

The authors used CESM model to perform idealized CO₂ ramp up and ramp down simulations in which atmospheric CO₂ increases at a rate of 1% per year to 4xCO₂, and then decreases at the same rate to 1xCO₂ (present-day level). The authors focused on the response of El Niño and the analyzed the role of deep ocean warming. By comparing El Niño events with present-day, the authors found that during the restoring period when atmospheric CO₂ returns to present-day level, Eastern Pacific El Niño events are intensified, precipitation in the tropical eastern Pacific increases, and convective extreme El Niño events also increase. By designing deep ocean warming simulations, the authors showed the importance of deep ocean warming in regulating El Niño response. This paper is well written. The simulations are well designed and the analysis is well done. I recommend its publication with minor modification.

We thank the reviewer for encouraging comments. The comments were fully considered in the revised manuscript. The point-by-point responses are as follows:

Comments:

1. How much heat is stored in the deep ocean and the resulting effect on El Niño response would depend on the rate of atmospheric CO₂ decrease. The author only performed ramp down simulations with a CO₂ decreasing rate of 1% per year. The implication of different CO₂ ramp down rate on the presented results should be discussed.

Ans) Thanks for asking an important question. Based on the reviewer's comments, we've added the discussion as follows:

The hysteresis of the precipitation and SST remains over the tropical Pacific even with a rapid CO₂ reduction, indicating that changes in ENSO are inevitable (Kim et al. 2022). The strength of the irreversibility depends on the rate of CO₂ reduction, which contributes to the amount of energy stored in the ocean. The slower the rate of reduction, the more heat will be stored in the ocean and the greater the warming of SST when CO₂ returns to current levels (Jeltsch-Thömmes et al. 2020). The results from the IW experiments show that the stronger the El Niño-like warming, the greater the increase in SST variability over the eastern Pacific (Fig. 2a). These results indicate that a slower rate of decline will result in a more prolonged and

pronounced warming of the tropical Pacific SST, which may potentially lead to a more intense El Niño event.

Specific comments:

Line 49: What results does 'here' refer to?

Ans) Revised to “This irreversible warming of the SST primarily occurs in the high latitudes and the equatorial eastern Pacific, where the oceans are weakly stratified to allow vigorous heat release through both vertical upwelling and mixing from the deep ocean to the surface.”

Lines 97-98: Please elaborate a bit more here of how the shift of the Pacific ITCZ partly contributes to the El Niño-like warming.

Ans) Revised “The southward shift of the Pacific ITCZ is accompanied by an increase in precipitation and a weakening of the trade winds over the equator, which further amplifies El Niño-like warming through the Bjerknes feedback.”

Lines 183-184: It might also be useful to show the vertical density gradient.

Ans) As the reviewer suggested, we have added the changes in density and vertical density gradient in supplementary Fig. 10 (Fig A).

Fig. A | The changes in ocean stratification. The difference in the equatorial averaged (5°S – 5°N) vertical climatological temperature and potential density from the PD period to (a) restoring, (b) IW_be700, (c) IW_be100 and (d) IW_whole. The difference in the equatorial averaged (5°S – 5°N) vertical gradient of climatological temperature and potential density from the PD period to (e) restoring, (f) IW_be700, (g) IW_be100 and (h) IW_whole. The shading, contour, and dashed lines indicate the temperature, potential density, and depth of the thermocline, respectively, in each experiment. Temperature values that are significant at the 95% confidence level using the bootstrap test are shown.

Lines 234-235: ‘These changes are driven by ocean stratification and El Niño-like warming, ‘. It appears twice.

Ans) Revised sentence to “These changes are driven by ocean stratification and El Niño-like warming, which enhance zonal advective, Ekman, and thermocline feedbacks in the eastern Pacific, despite increased thermal damping (Supplementary Fig. 13). Our findings consistently show that deep ocean warming is the primary driver of these ENSO changes, as observed in the IW_EXPs.”

Line 288: How long does the current-day simulation last?

Ans) The present-day (PD) simulation was performed for a total of 900 years, and the PD period used in the analysis is defined as the last 400 years of the PD simulation. We revised the sentence to “The first simulation was a present-day scenario with a constant atmospheric CO₂ concentration (at the current level of 367 ppm) for 900 years.”

Line 309: How is the salinity anomaly profile determined?

Ans) The salinity anomaly was calculated using the same method as used for the temperature anomaly. We revised the sentence to “Additionally, horizontally uniform vertical profiles of the ocean temperature and salinity anomalies were added to the initial condition of the ocean model (year 2000). In other words, the initial temperature and salinity perturbations of IW_EXPs are horizontally identical in all ocean grids.”

Lines 379-380: Deep ocean warming induced irreversible climate changes Nature Climate Change, Nature Climate Change. For some reason, I could not find this article.

Ans) The information on the reference has been updated.

Oh, J.-H. *et al.* Emergent climate change patterns originating from deep ocean warming in climate mitigation scenarios. *Nat. Clim. Chang.* **14**, 260–266 (2024).

Fig. 1 caption (b) and (c) should exchange with each other. Or, the numbering of each panel should be changed.

Ans) The Fig. 1 caption has been revised accordingly.

Response to Reviewer #2

Kim et al. investigated the variations in ENSO during the recovery period after intensive global warming. Generally, this study showed some interesting results helping us to deepening understanding of ENSO, based on a series of numerical experiments. But I still have some concerns, which are listed below.

We would like to thank the reviewer for careful and constructive suggestions, which helped to improve the quality of this manuscript. Our responses are as follows:

1. The mitigation scenario in the title is somewhat unrelated to this study.

Ans) Thanks for pointing out this issue. As the reviewer suggested, we have revised the title of the paper to "Deep ocean warming-induced El Niño changes".

2. "As the climate recovers after CO₂ reaches current climate levels, the Northern Hemisphere (NH) cools much faster than the Southern Hemisphere (SH)". (1) The authors should give evidence to support this. (2) What causes the different cooling rates? (3) And what is the relative contribution of the NH-SH contrast and the El Niño-like warming pattern to the migration of the ITCZ?

Ans for (1) and (2): Figure B shows the difference in the air temperature between the PD period and the restoring period. The temperature anomaly is higher in the Southern Hemisphere (SH) than in the Northern Hemisphere (NH), indicating that the SH cools more slowly. Kug et al. (2022) pointed out that the cooling of the SH during climate recovery is significantly delayed by its heat capacity associated with the land fraction. In addition, Oh et al. (2024) suggested that heat release from the deep ocean is stronger in the Southern Ocean, a weakly stratified background condition, which contributes to the slower cooling of the SH during climate recovery. We have added the supplementary Figure and revised the manuscript as follows :

“As the climate recovers after CO₂ reaches current levels, the Northern Hemisphere (NH) cools much faster than the Southern Hemisphere (SH) (Supplementary Fig. 2). The larger heat

capacity of the SH (Kug et al. 2022) and the pronounced heat release from the Southern Ocean (Oh et al. 2024) contribute to a relatively slow cooling rate in the SH.”

Fig. B | The changes in the air temperature. The difference in (a) the spatial pattern and (b) the zonal averaged the air temperature from the PD period to restoring. The air temperature is vertically averaged from the surface to 300hPa.

Ans for (3): It is a bit difficult to quantify the relative contribution of the SH-NH contrast and the El Niño-like warming to the southward shift of the ITCZ, since they are closely related. However, we tried to examine the relative importance based on the inter-ensemble spread. The shift of the ITCZ index was defined as the difference in precipitation between the northern (15°-5°N, 120°E-120°W) and southern (15°-5°S, 120°E-120°W) off-equatorial regions. El Niño-like warming was defined as the mean SST in the Niño3 region (5°S-5°N, 150-90°W). The inter-hemispheric temperature contrast is defined as the difference between the tropics (0°-90°N and 90S°-0°) of the NH and SH annual mean temperature, which is vertically averaged from the surface to 300hPa. Figure C shows the inter-model relationship between El Niño-like warming, interhemispheric contrast, and the ITCZ shift. El Niño-like warming and interhemispheric contrast correlated with the southward shift of the ITCZ by 0.68 and 0.41, respectively. The results indicate that a stronger inter-hemispheric contrast and a stronger El

Niño-like warming are both associated with a larger southward shift of the ITCZ. In order to compare the relative contribution of the SH-NH contrast and El Niño-like warming, partial correlation analysis is employed. The partial correlation coefficient $r_{12.3}$ between two variables A_1 and A_2 , after removing the influence of the variable A_3 , is given by:

$$r_{12.3} = \frac{r_{12} - r_{12}r_{23}}{\sqrt{(1 - r_{13}^2)(1 - r_{23}^2)}}$$

The coefficients for El Niño-like warming and SH-NH contrast are 0.64 and 0.39, respectively, suggesting that El Niño-like warming is more associated with the southward shift of the ITCZ. A more accurate quantitative assessment of the relative roles of the two factors will require further modeling experiments and will be a very interesting topic for future research.

Fig. C | The relationship between El Niño-like warming, interhemispheric contrast, and the shift of the ITCZ. The scatter diagram (a) between the El Niño-like warming and the shift of the ITCZ and (b) between the interhemispheric contrast and the shift of ITCZ.

3. “The initial warming over the eastern Pacific “in line 121. The authors may show the initial warming pattern in supplementary material, which is very helpful to understanding the formation processes of the El Niño-like warming pattern.

Ans) To represent the initial SST warming pattern, we calculated for each experiment the SST difference between the PD period and the first 5 years of the experiment (Fig. D). Based on the reviewer's comment, we have added the initial warming pattern to Supplementary Fig. 4

Fig. D | The initial warming pattern of the IW experiment. The difference in SST from the PD period to the initial five-year of (a) IW_be700, (b) IW_be100, and (c) IW_whole.

4. The authors stressed the intensification of the zonal-advection feedback. However, in previous studies, this feedback favors the CP El Niños?

Ans) As the reviewer mentioned, the zonal advection feedback is of critical importance in the CP El Niño, while the thermocline feedback represents a key process in the EP El Niño (Kug et al. 2009). However, in terms of changes in ENSO diversity, the region where the feedbacks change is more important. In response to El Niño-like warming, the wind response shifts eastward. As a result, the ocean current response intensifies in the eastern Pacific (150-120W) rather than in the central Pacific in the restoring and all IW_EXPs. (Fig E). In addition to zonal currents, vertical currents, and thermocline responses are enhanced in the eastern Pacific (Supplementary Fig. 11). Therefore, the frequency of EP El Niños increases as zonal advection in the eastern Pacific intensifies.

Fig. E | The changes in the current response. The difference of the regression coefficients of zonal current onto averaged wind stress over the equatorial Pacific (120°E–90°W, 5°S–5°N) during DJF from the PD period to **(a)** restoring, **(b)** IW_be700, **(c)** IW_be100 and **(d)** IW_whole. Only values that are significant at the 95% confidence level using the bootstrap test are shown.

5. “The second experiment lasted 500 years and involved 28 ensemble members.” Do you mean 500*28 years of integrations were conducted?

Ans) Yes. The sentence is revised as follows:

“The second experiment consists of 28 ensemble members with different initial conditions, for a total of 500 years in each ensemble.”

6. *“A horizontally uniform temperature anomaly profile was added to the initial ocean state”.*
Why uniform temperature anomalies are specified? This should be very different from those in recovery experiments.

Ans) Our intention of these experiments is to emphasize that the climatological distribution of the oceanic vertical mixing is a key in determining the deep ocean warming-induced climate changes. As the reviewer pointed out, the added globally-uniform ocean warming pattern is quite different from the actual distribution of the deep ocean warming in the restoring period. Nevertheless, the simulated tropical mean states are almost identical to those in the restoring period. This suggests that the climatological equatorial upwelling is a key to transferring the heat from the deep ocean, regardless of the horizontal distribution of the deep ocean warming.

7. *Various feedbacks were diagnosed, but the authors did not mention which seasons they analyzed*

Ans) A description of the seasons was added for each Figure and supplementary caption.

Response to Reviewer #3

This study explored ENSO changes due to deep-ocean warming based on a CO₂ ramp-up and ramp-down experiment and present-day experiments initialized with horizontally uniform deep-ocean heat anomalies using CESM1.2. The authors examined not only the ENSO amplitude changes but also the diversity and feedback changes.

I have read several previous papers related to the CESM1.2 CO₂ ramp-up/down experiments. This paper aims to clarify the impact of the deep-ocean heat on the hysteresis of ENSO, which is important to the ENSO community. Although the results are interesting, I have some potentially critical concerns in their analyzing method. Because the present manuscript does not sufficiently describe their method, I could not fully understand how the data was analyzed. I would like to ask the authors to revise the methodology and clarify my major concerns below.

We appreciate the reviewer's careful and constructive comments. The reviewer's comments were fully reflected in the revised manuscript, and we carefully checked and revised the manuscript. Our responses to the specific comments are as follows:

Major comments:

1. The IW experiments are not at the equilibrium states, so the anomalous fields have to be carefully defined to compare with the PD experiment. The last 100 years after 50 years are analyzed in the IW experiments and the PD period (Lines 106-107). However, it is clear in Fig. 3b of Oh et al. (2024), that the SST change is not steady at all. Thus, all the anomalous fields have to be carefully defined.

As the mean state remains changing for 150 years, it is important to describe it and consider removing its potential impact on the anomalous fields analyzed in this study. First of all, the definition of anomalies needs to be clarified. Then, how to remove the potential background changes from the anomalies needs to be described. For example, the Niño3 SSTA std. is easy to be affected by spurious slow anomalies due to background change. The SSTA pattern in Fig. 2 is also affected by the definition. Other main and supplementary figures are also sensitive to

the definition of the anomalies.

Ideally, an initial condition large ensemble is required to define the anomalies from the changing background, but the experiments have only 9 ensembles for each of the IW experiments. Therefore, the method to define the anomaly is important.

Ans) We agree with the reviewer that defining the anomalies can be very important. Given the limited number of ensembles in the IW experiments, it is not appropriate that anomalies are defined by removing the ensemble mean to remove the inherent changing background. Therefore, anomalies of each ensemble were defined by removing the linear trend and the monthly mean climatology for the last 100 years.

Furthermore, we also checked the sensitivity of our results to the definitions of anomalies. We conducted the composite analysis for the El Niño using anomalies defined as removing only the monthly mean not the linear trend. Figure F shows the SSTA difference from the PD period. The SSTA difference shows a zonal dipole pattern, showing that the equatorial eastern and western Pacific regions have opposite signs. These results are consistent with those obtained by removing the linear trend (Fig. 2b,c in the manuscript), indicating that the impact of spurious slow anomalies due to background change on the ENSO composite result is very limited.

Following the reviewer's comment, how we defined anomalies is added in the method section as follows:

Definition of ENSO

The Niño3 and Niño4 SSTA, defined as the area-averaged SSTA in the Niño3 region (5°S-5°N, 150-90°W) and Niño4 region (5°S-5°N, 160°E-150°W) are used to describe the amplitude of ENSO, respectively. The SSTA was calculated by removing the linear trend and the monthly climatological mean for each experiment and ensemble. We utilize a threshold of ± 1 STD of the DJF mean Niño3 SSTA during the PD period to define ENSO events. To define convective El Niño, we employed the method used in previous studies (Cai et al. 2014, 2015; Wang et al. 2017; Pathirana et al. 2023), wherein convective extreme El Niño was identified as absolute precipitation exceeding a threshold of 5 mm/day during the DJF season.

Fig. F | The changes in ENSO amplitude. The changes in the spatial pattern of El Niño SSTA during DJF from the PD period (a) IW_be700, (b) IW_be100, and (c) IW_whole. El Niño was defined using the 1 STD threshold of the DJF Niño SSTA. Stippled areas are regions that are not significant at the 95% differences of each experiment using the bootstrap test.

2. The IW experiments are not fully described (Lines 104-105 and the Methods section), so it is not possible to reproduce the simulations based on the manuscript. (1) The method could be explained more in detail. At least, it must be clarified if the experiments are identical to the previous work (ref. 17).

Ans for (1): Thanks for asking the important point. The experiments are identical to those in Oh et al. (2024). We have described the design of the IW experiments in the Method sections

in more detail as follows:

Results

To investigate the influence of deep ocean warming on changes in the tropical mean state, three initial warming experiments (IW_EXPs) were conducted. Specifically, the IW_EXPs were conducted with a constant atmospheric CO₂ level, but with horizontally uniform vertical profiles of ocean temperature and salinity anomalies added to the ocean model initial conditions. These IW_EXPs allow us to explore the impacts of deep ocean warming on the mean state of the tropical ocean and subsequent alterations in ENSO characteristics (see Methods for a detailed description of the IW_EXPs).

Design of the initial warming experiment

To investigate how deep ocean warming contributes to the change in ENSO characteristics, three types of initial warming experiments (denoted as IW_whole, IW_be100, and IW_be700) were conducted. These experiments were branched from the initial condition (year 2000) of each ensemble member and then integrated with the constant atmospheric CO₂ concentration of 367 ppm. Additionally, horizontally uniform vertical profiles of the ocean temperature and salinity anomalies were added to the initial condition of the ocean model (year 2000). In other words, the initial temperature and salinity perturbations of IW_EXPs are horizontally identical in all ocean grids. The added initial anomalies are the global mean ocean temperature and salinity in the year 2280 when the atmospheric CO₂ level first returns to its original level (367 ppm). Note that, in addition to the temperature profile, a salinity anomaly profile was added to prevent any potential imbalance for running the model. To isolate the role of deeper ocean warming, the initial perturbation was added to the entire depth, below 100m and below 700m in IW_whole, IW_be100, and IW_be700, respectively. All experiments were integrated for 150 years with nine ensemble members. Note that these three IW_EXPs were also used in a previous study (Oh et al. 2024, see their Fig. 3).

And, (2) what are the values of the uniform temperature anomaly profile and the corresponding salinity anomaly profile? (3) How did you determine the salinity anomaly profile to avoid any potential density imbalance?

Ans for (2) and (3): As described in the Methods section, the added initial vertical profile is the global mean ocean temperature and salinity anomalies in the year 2280 when the atmospheric CO₂ concentration first returns to its original level (367 ppm, Fig. G and H). To avoid any potential imbalance in running the model, we simply added the global mean salinity profile in addition to the temperature profile.

To compare the sensitivity of our results to the salinity, We conducted additional IW_whole experiments, which separately added the temperature and salinity anomalies to the initial ocean state. Figure I shows the change in SST from the PD period to the IW_whole experiment with temperature and salinity added respectively. The experiment where only the temperature was added shows a pronounced warming at the surface (Fig. Ia), with an El Niño-like warming pattern, consistent with the experiment where both temperature and salinity were added (Fig. 1 in the manuscript). In contrast, the experiment where only salinity was added shows no clear surface warming pattern (Fig. Ib). These results suggest that how to deal with the salinity anomalies does not change our main conclusion significantly.

Fig. G | Idealized CO₂ scenario imposed in the Earth System Model experiment. The black line indicates the CO₂ pathway with model year. Present day (PD, 1100-2000 years), ramp-up (2000-2140 years), ramp-down (2140-2280 years), and restoring period (2280-2500 years) were delineated on the figure.

Fig. H | Design and results of the initial warming experiment (Adopted from Fig 3a in Oh et al, 2024). Global mean potential temperature (red) and salinity (black) profile in year 2280 when the CO₂ concentration first returns to the present level. These two vertical profiles are uniformly added to all ocean grids of the model's original initial condition (year 2000 condition). The two grey horizontal lines denote the 100 m and 700 m depths, respectively. The red, blue, and yellow vertical bars indicate the depth of the initial forcing used in each experiment.

Fig. 1 | The changes in SST. The changes in the spatial pattern of SSTA from PD to IW_{whole} experiments. **(a)** Only horizontally uniform temperature anomaly profiles and **(b)** only salinity anomaly profiles were added to the initial ocean state. The SSTA was defined as the difference between the PD period and the time average of the last 100 years of the simulation.

(4) Is there an initial condition's vertical gap at the 700-m (100-m) depth in IW_{be700} (IW_{be100})?

Ans for (4): In the IW_{be100} and IW_{be700}, the initial perturbation was set to zero for the upper 100m and 700m, respectively. Therefore, as the reviewer asked, there exists a vertical discontinuity of initial perturbation at the 100m and 700m in the IW_{be100} and IW_{be700}, respectively. We acknowledge that this discontinuity can be somewhat unrealistic. However, it seems that it may not significantly impact our overall findings, because the results are quite similar to those in the IW_{whole}, which utilized a complete vertical anomaly profile and yielded similar results.

(5) The total heat content from the bottom to the surface may depend on the bottom topography, so probably the total heat content is not horizontally uniform --Does this affect the results? Because these points were unclearly written, I could not follow the detailed parts of the results.

Ans for (5): As the reviewer points out, the ocean heat content (OHC) from the bottom to the surface is not horizontally uniform due to the topography of the ocean. To investigate the effect of differences in the OHC as the topographic properties of the oceans, the OHC added as the

initial forcing was divided into the above and the below 3000m (Fig. Ja, b). The OHC integrated over 3000 m shows an almost identical value over the globe, except along coastlines with low water depths. On the other hand, the ocean heat content integrated from 3000 m to the bottom is not spatially uniform, with positive values in regions of deep ocean depth, such as the northwest Pacific. To make a quantitative comparison, the ratio was defined as the OHC below 3000m relative to the OHC above 3000m (Fig. Jc). The values are between 0.1 and -0.1 in all regions, indicating that the amount of OHC below 3000 meters is relatively very small. Furthermore, its spatial pattern is not associated with SST warming (Fig.1 in the manuscript), with a pattern correlation coefficient of -0.06, which is not statistically significant. Therefore, it seems that the regional difference in OHC added to the initial condition due to ocean topography may not significantly impact our overall findings.

Fig. J | The ocean heat content. The ocean heat content (OHC) integrated from (a) the surface to 3000m and (b) 3000m to the bottom. (c) The ratio of OHC above 3000m to below 3000m. The OHC was calculated using the temperature anomalies which were added to the initial condition. The shaded unit (a) is 10 times larger than (b).

3. Adding a schematic chart to explain the present conclusion may help readers understand what is different between the mechanism underlying ENSO changes induced by deep ocean warming and that associated with anthropogenic greenhouse warming (Lines 243-252).

Ans) Thank you for the constructive suggestions. Figure K shows the schematic chart for the mechanism of the changes in ENSO induced by greenhouse and deep ocean warming. In the second step, the process that is more dominant is indicated by a greater-than symbol, while the less dominant process is stippled. As the reviewer suggested, we have added a schematic chart to the supplementary Fig. 14 (Fig. K).

Fig. K | The schematic chart for the mechanism of the changes in ENSO. Mechanisms for El Niño changes induced by (a) the greenhouse and (b) deep ocean warming. The dominant process is indicated by a greater-than symbol, while the other is stippled. Greenhouse gas warming increases ocean surface temperatures, leading to increased ocean stratification, and consequently intensifies the ocean response in the central Pacific, promoting more frequent CP-type El Niño events. Deep ocean warming begins from a state of initially reduced stratification, limiting its stratification effect. El Niño-like warming response to deep ocean warming contributes to an eastward shift in the wind response, which reinforces the ocean response in the eastern Pacific, culminating in a higher frequency of EP-type El Niño events.

4. The criteria for defining ENSO are not clear to me.

In Line 135, the authors said that "El Niño events were defined based on 1 st.d. of the Niño3 SST." Are these criteria the same among the experiments? Or, did you use different thresholds in each experiment? This difference may affect the patterns of the SSTA in Fig. 2c,d,e., for example.

Ans) In the previous manuscript, the different thresholds were applied by calculating the standard deviation for each experiment and ensemble (Fig. L). As in response to comment #1, we have added a definition for El Niño events to the methods. In order to ascertain the changes in the SSTA patterns in accordance with the thresholds, the composite analysis for the El Niño was conducted using identical thresholds for all experiments. El Niño was defined using the 1 standard deviation threshold of the DJF Niño SST in the PD period (Fig. M). The SSTA difference shows a zonal dipole pattern, suggesting that the variability of SST in the eastern Pacific is increasing. These results are consistent with those obtained by using different thresholds for each ensemble (Fig. 2b,c in the previous manuscript). This implies that employing a distinct threshold does not significantly alter our main conclusions. Based on these results, the manuscript and figures have been fully replaced by applying identical thresholds in all experiments when defining El Niño (Fig N and O).

Fig. L | The thresholds of ENSO. The standard deviation of DJF Niño3 SSTa for each ensemble in each experiment and ensemble.

Fig. M | The changes in ENSO amplitude. The changes in the spatial pattern of El Niño SSTA during DJF from the PD period **(a)** restoring, **(b)** IW_be700, **(c)** IW_be100 and **(d)** IW_whole. El Niño was defined using the 1 STD threshold of the DJF Niño SSTA in the PD period. Stippled areas are regions that are not significant at the 95% differences of each experiment using the bootstrap test.

Fig. N | The changes in ENSO amplitude. The standard deviation of the DJF **(a)** Niño3 and **(b)** Niño4 SSTA. **(c)** The spatial pattern of El Niño SSTA during DJF for the PD period. El Niño was defined using the 1 STD threshold of the DJF Niño SSTA. As in **(c)** but for the difference from the PD period to **(d)** restoring and **(e)** IW_be700. Error bars represent the 95% confidence interval using the bootstrap test. Stippled areas are regions that are not significant at the 95% confidence level of the Student's t-test in the PD period and for the differences of restoring and IW_be700 using the bootstrap test.

Fig. O | The changes in ENSO's flavor. (a) Histogram (normalized occurrences) of El-Niño SSTA centers during DJF for the PD period. As in (a) but for the difference from the PD period to (b) restoring and (c) IW_be700. (d) Ratios of frequencies of CP El Niño events proportion to the total number of CP and EP El Niño events. (e) The frequency of convective El Niño defined as absolute precipitation exceeding a threshold of 5 mm/day during the DJF season. Error bars represent the 95% confidence interval using the bootstrap test.

5. The convective El Niño is not clearly defined in the text.

In Line 166, the authors described "the convective El Niño was defined as when Niño3 rainfall during the DJF season exceeds a threshold of 5 mm/day". Do the authors mean "Niño3 precipitation anomaly" instead of "the absolute precipitation"? If the absolute precipitation in the Niño3 region is used for defining the convective El Niño, the increase of the convective El Niño event occurrences is attributed to the mean state change to a certain extent.

Ans) For convective extreme El Niño, we just adopted the definition used in previous studies (Cai et al. 2014, 2015; Wang et al. 2017; Pathirana et al. 2023), wherein convective extreme El Niño was identified as absolute precipitation exceeding a threshold of 5 mm/day during the DJF season. As in response to comment #1, we have added a definition for El Niño events to the methods.

As the reviewer points out, the increase in convective El Niño events is to some extent due to the change in mean state. However, the increase in extreme El Niño events is not simply due to an increase in climatological precipitation, but to an increase in the probability of deep atmospheric convection in the eastern equatorial Pacific due to the changes in the mean state change.

6. It is unclear if there is inter-model uncertainty in the ENSO changes during the restoring period.

In the Introduction, the authors imply that there are multi-model datasets of CDRMIP, which can be used to investigate the inter-model uncertainty of the present key results. I would suggest the authors conduct a multi-model analysis to indicate whether the major findings are independent of the models or dependent on the models.

Ans) Based on the reviewer's comments, we analyzed the CDRMIP data and added the results to the revised manuscript and supplementary.

To strengthen the robustness of our findings, we further analyzed the 8 model simulations from the Carbon Dioxide Removal Model Intercomparison Project (CDRMIP)(Keller et al. 2018) of CMIP6, which has a similar experimental protocol to the simulation in this study. All of the model data are bilinearly interpolated onto a common $1^\circ \times 1^\circ$ grid before analysis. The length of the recovery period (constant CO₂ forcing period) varies among models (Table A). Note that several models provide too short restoring period to quantify significant El Niño changes.

Figure Pa shows the difference in the temperature from the picontrol to the restoring period. The SST pattern in the MME is quite similar to the result of this study, implying that the deep ocean warming induces the El Niño-like warming in the restoring period. To check the changes in ENSO characteristics, we calculate the changes in the STD of the Niño3 SSTA from the

picontrol to the restoring period (Fig Pb). Five of the eight models show an increase in the STD of the Niño3 SSTA, with four showing an increase of more than 8%. GFDL-ESM4, CESM2, and CNRM-ESM2-1 show relatively large increases, but these are not statistically significant, due to the short recovery period (Table A). The MME values show a statistically significant increase of about 7%. Convective extreme El Niño increased in all models except the MIROC-ES2L model, although CanESM5 has no value as no convective El Niño occurred in either period (Fig Pc). In particular, the top four models with large increases in the Niño3 STD show a large increase in the frequency of convective El Niño events. The MME results indicate that the number of convective extreme El Niño events has been about double during the restoring period. The CDRMIP results are overall consistent with our main results, suggesting the robustness of our results, but caution is needed for the short restoring period.

Fig. P | The changes in the tropical mean state and ENSO amplitude in CMIP6. (a) The difference in SST between the restoring and PD period. The difference in (b) the standard deviation of the DJF Niño3 SSTA and (c) the frequency of convective El Niño between the restoring and PD period. Error bars represent the 90% confidence interval using the bootstrap test. Stippled areas are regions that are not significant at the 90% confidence level for the differences of restoring and PD period using the bootstrap test.

Table A | CMIP6 8 models list.

Model number	Model name	Model year (pi-control)	Model year (restoring)
1	ACCESS-ESM1-5	1000	620
2	CESM2	1200	60
3	CNRM-ESM2-1	500	60
4	CanESM5	1051	60
5	GFDL-ESM4	500	60
6	MIROC-ES2L	500	362
7	NorESM2-LM	501	119
8	UKESM1-0-LL	1880	510

Minor comments:

Title: I would say that this study analyzed El Niño changes under a single climate mitigation scenario in a climate model, so saying "under climate mitigation scenarios" does not seem representative of the study.

Ans) We agree with the reviewer, so we have revised the title of the paper to "Deep ocean warming-induced El Niño changes".

Lines 24-25: This sentence is too subjective and ambiguous to describe the implications of this study.

Ans) Revised the sentence to “Our findings suggest that anthropogenic greenhouse warming will have a prolonged impact on El Niño variability through delayed deep ocean warming, even if CO₂ stabilization is achieved.”

Line 51: The meaning of "residual upwelling" is unclear.

Ans) Revised to “This irreversible warming of the SST primarily occurs in the high latitudes and the equatorial eastern Pacific, where the oceans are weakly stratified to allow vigorous heat release through both vertical upwelling and mixing from the deep ocean to the surface.”

Line 64: I am quite unsure of the reason for citing ref. 28 (Ohba and Ueda 2009, JCLI <https://doi.org/10.1175/2008JCLI2334.1>).

Ans) Removed

Line 79: What is "the model"? The details can be found in the Methods section, but the model name could be provided in the main text for clarity.

Ans) Revised the manuscript to “The Community Earth System Model (CESM) was run with constant atmospheric CO₂ (restoring) after it was increased for 140 years, then symmetrically

decreased (see Methods and Supplementary Fig. 1).”

Line 129: Actually, the Niño4 SSTA st.d. is decreased in IW_whole. How did you interpret this?

Ans) Thanks for asking us an interesting point. However, it is difficult to address this issue accurately. The IW_whole also shows the largest increase in Niño3 SST. Actually, the differences from the other experiments are not statistically significant, we may need longer-term simulation to explore this issue further.

Line 130: The Niño3 SSTA st.d. is increased more in IW_be100 and IW_whole than in IW_be700 and Restoring. Why this happened could be explained in the text.

Ans) Figure Q shows the change in SST from the PD period to the restoring period and to all IW_EXPs. The larger increase in the STD of the Niño3 SSTA is probably due to the more pronounced El Niño-like warming in IW_be100 and IW_whole. We have revised the sentence as follows: The largest increases are in the order of IW_whole, IW_be100, and IW_be700, which are proportional to the tropical mean state warming (Supplementary Fig. 3).

Fig. Q | The changes in SST. The difference in SST from the PD period to (a) restoring, (b) IW_be700, (c) IW_be100 and (d) IW_whole. Only significant values at the 95% confidence level using the bootstrap test are shown.

Line 136-137: How to calculate the central longitude is unclear. The Method section needs to be cited here.

Ans) It simply detected the longitude at which the maximum value occurs.

Line 137: Any figure does not describe this text. A supplementary figure needs to be provided.

Ans) Figure R shows the spatial pattern of El Niño SSTA and its equatorial mean (5°S-5°N) in the recovery period and all IW_EXPs. We have added Figure R to the supplementary Fig. 5.

Fig. R | The spatial pattern of El Niño. The spatial pattern of El Niño SSTA (shading) during DJF and its equatorial average (5°S-5°N, red line) in (a) restoring, (b) IW_700 (c) IW_be100 and (d) IW_whole. Stippled areas are regions that are not significant at the 95% confidence level of Student's t-test.

Lines 150-151: The observational distribution is not shown in any figure.

Ans) We added the reference and revised the sentence as follows: The zonal distribution of SSTA peaks shows a bimodal structure with maxima at 155W and 115W (Fig. 3a), although the model tends to simulate more CP El Niño compared to the observational distribution (Fig. 1a-b in Shin et al. 2022).

Figure 4: Is the figure calculated based on DJF? Please clarify the season used for this figure.

Ans) Added the season to the caption.

Line 190: Where is the thermocline in Fig. 4a-c?

Ans) Added the thermocline depth in Fig. 4a-c.

Line 213: Define the Niño3 index. Do you mean Niño3 SSTA? In this manuscript, the index names are confusingly mixed (e.g., Niño3, Niño3 SST, Niño3 SSTA, Niño3 index).

Ans) Revised to “Niño3 SSTA”. In addition, we have unified the expressions “SSTA” and “Niño3 SSTA” throughout the manuscript.

Line 220: The results from the IW_be700 experiment are not mentioned yet. Figure 4f should appear in Lines 221-222.

Ans) Revised.

Figure 2a,b: What is the season for the Niño3 and Niño4 SSTA standard deviations? Write in the caption.

Ans) Added the season to the caption.

Typos etc.:

Line 21: Use "Intertropical Convergence Zone" or "intertropical convergence zone" instead of "Intertropical convergence zone".

Ans) Revised to “Intertropical Convergence Zone”.

Line 84: Do you mean "The present-day period" instead of "the restoring period"?

Ans) Revised.

Line 129, etc.: Use "Niño3 SSTA" instead of "Niño3 SST"? Check the manuscript overall.

Ans) Revised to “Niño3 SSTA”. In addition, we have unified the expressions “SSTA” and “Niño3 SSTA” throughout the manuscript.

Line 130, 141, 496, etc.: Use "Niño3 SSTA" instead of "Niño3"? Check the manuscript overall.

Ans) Revised.

Line 188 etc.: Use "wind stress" instead of "windstress" overall.

Ans) Overall, revised every instance of 'windstress' to 'wind stress' in the manuscript.

Line 210: Replace "Fig" with "Fig.".

Ans) Revised

Lines 235-236: This sentence repeats the previous sentence in Lines 234-235.

Ans) Revised sentence to “These changes are driven by ocean stratification and El Niño-like warming, which enhance zonal advective, Ekman, and thermocline feedbacks in the eastern Pacific, despite increased thermal damping (Supplementary Fig. 13). Our findings consistently show that deep ocean warming is the primary driver of these ENSO changes, as observed in the IW_EXPs.”

Line 272: "STD" is not defined.

Ans) Revised the manuscript (L135-136) to define the STD as follows: The standard deviation (STD) of the Niño3 SSTA increases significantly, while the Niño4 SSTA remains unchanged except for IW_whole.

Lines 331, 361: Check the titles of these references. "CO2" instead of "CO 2". And, "2°C" instead of "2 C".

Ans) Revised the reference.

References (e.g., ref. 14): The authors' names are not consistent in the references. for example, JS, or J-S? Check all references for consistency.

Ans) Revised the format of references to be consistent.

Reference 17: Replace "cPhaden" with "McPhaden".

Ans) Revised the reference.

Line 380: Add a period at the end of the title.

Ans) Revised the reference.

Reference 17: I could not find ref. 17. Has it changed from the cited one to the following published paper? Oh, JH., Kug, JS., An, SI. et al. Emergent climate change patterns originating from deep ocean warming in climate mitigation scenarios. Nat. Clim. Chang. (2024). <https://doi.org/10.1038/s41558-024-01928-0>.

Ans) Revised the reference.

Reference 31: I could not reach this reference (Pathirana et al. 2022). Is the reference changed from the cited one to the following paper? Gayan Pathirana et al. Increase in convective extreme El Niño events in a CO2 removal scenario. Sci. Adv. 9, eadh2412 (2023). DOI:10.1126/sciadv.adh2412

Ans) Revised the reference.

Reference

- Cai, W., and Coauthors, 2014: Increasing frequency of extreme El Niño events due to greenhouse warming. *Nature Clim Change*, **4**, 111–116, <https://doi.org/10.1038/nclimate2100>.
- , and Coauthors, 2015: ENSO and greenhouse warming. *Nature Clim Change*, **5**, 849–859, <https://doi.org/10.1038/nclimate2743>.
- , G. Wang, A. Santoso, X. Lin, and L. Wu, 2017: Definition of Extreme El Niño and Its Impact on Projected Increase in Extreme El Niño Frequency. *Geophysical Research Letters*, **44**, 11,184–11,190, <https://doi.org/10.1002/2017GL075635>.
- Jeltsch-Thömmes, A., T. F. Stocker, and F. Joos, 2020: Hysteresis of the Earth system under positive and negative CO₂ emissions. *Environ. Res. Lett.*, **15**, 124026, <https://doi.org/10.1088/1748-9326/abc4af>.
- Keller, D. P., and Coauthors, 2018: The Carbon Dioxide Removal Model Intercomparison Project (CDRMIP): rationale and experimental protocol for CMIP6. *Geoscientific Model Development*, **11**, 1133–1160, <https://doi.org/10.5194/gmd-11-1133-2018>.
- Kim, S.-K., J. Shin, S.-I. An, H.-J. Kim, N. Im, S.-P. Xie, J.-S. Kug, and S.-W. Yeh, 2022: Widespread irreversible changes in surface temperature and precipitation in response to CO₂ forcing. *Nat. Clim. Chang.*, **12**, 834–840, <https://doi.org/10.1038/s41558-022-01452-z>.
- Kug, J.-S., F.-F. Jin, and S.-I. An, 2009: Two Types of El Niño Events: Cold Tongue El Niño and Warm Pool El Niño. *Journal of Climate*, **22**, 1499–1515, <https://doi.org/10.1175/2008JCLI2624.1>.
- , and Coauthors, 2022: Hysteresis of the intertropical convergence zone to CO₂ forcing. *Nat. Clim. Chang.*, **12**, 47–53, <https://doi.org/10.1038/s41558-021-01211-6>.
- Oh, J.-H., J.-S. Kug, S.-I. An, F.-F. Jin, M. J. McPhaden, and J. Shin, 2024: Emergent climate change patterns originating from deep ocean warming in climate mitigation scenarios. *Nat. Clim. Chang.*, **14**, 260–266, <https://doi.org/10.1038/s41558-024-01928-0>.
- Pathirana, G., J.-H. Oh, W. Cai, S.-I. An, S.-K. Min, S.-Y. Jo, J. Shin, and J.-S. Kug, 2023: Increase in convective extreme El Niño events in a CO₂ removal scenario. *Science Advances*, **9**, eadh2412, <https://doi.org/10.1126/sciadv.adh2412>.
- Wang, G., W. Cai, B. Gan, L. Wu, A. Santoso, X. Lin, Z. Chen, and M. J. McPhaden, 2017: Continued increase of extreme El Niño frequency long after 1.5 °C warming stabilization. *Nature Clim Change*, **7**, 568–572, <https://doi.org/10.1038/nclimate3351>.

REVIEWERS' COMMENTS

Reviewer #2 (Remarks to the Author):

The authors have solved my concerns satisfactorily. I have no further questions.

Reviewer #3 (Remarks to the Author):

May 2024,

The review of the revised manuscript "Deep Ocean warming-induced El Niño changes".

I fully appreciate the substantial efforts made by the authors in addressing my concerns. The revised manuscript now includes detailed descriptions of the methods, which has certainly improved the quality of the manuscript. In particular, the authors have changed the El Niño event detections and showed that the results were essentially unchanged. Although the El Niño SSTA differences shown in Fig. 2d and 2e are a bit reduced, these changes do not affect their main results (e.g., Fig. 3 and Fig. 4). The authors have also addressed my concerns about the initial warming profiles and the anomaly definition in their reply, which are supportive to their conclusions. I also found that their new results on a CMIP6 ensemble are consistent with their main conclusions from CESM1.2 simulations (Supplementary Note and Supplementary Fig. 6), showing the robustness of their original findings. Furthermore, their new schematic figure (Supplementary Figure 14) is useful to understand what they found.

I did not find any additional issues that need clarification, except for the changes in Fig. 4b and 4c from the previous version. I could not find the reason why the statistically significant areas in Fig. 4b and 4c became much wider compared to the previous Fig. 4b and 4c. Please confirm if the statistical test in Fig. 4b and 4c was conducted correctly.

Response to Reviewer #2

The authors have solved my concerns satisfactorily. I have no further questions..

We thank the reviewer for encouraging and constructive comments which allowed us to further develop the paper.

Response to Reviewer #3

I fully appreciate the substantial efforts made by the authors in addressing my concerns. The revised manuscript now includes detailed descriptions of the methods, which has certainly improved the quality of the manuscript. In particular, the authors have changed the El Nino event detections and showed that the results were essentially unchanged. Although the El Nino SSTA differences shown in Fig. 2d and 2e are a bit reduced, these changes do not affect their main results (e.g., Fig. 3 and Fig. 4). The authors have also addressed my concerns about the initial warming profiles and the anomaly definition in their reply, which are supportive to their conclusions. I also found that their new results on a CMIP6 ensemble are consistent with their main conclusions from CESM1.2 simulations (Supplementary Note and Supplementary Fig. 6), showing the robustness of their original findings. Furthermore, their new schematic figure (Supplementary Figure 14) is useful to understand what they found.

We would like to thank the reviewer for their thoughtful and constructive suggestions, which have helped to enhance the quality of this manuscript. The response is as follows:

Comment:

I did not find any additional issues that need clarification, except for the changes in Fig. 4b and 4c from the previous version. I could not find the reason why the statistically significant areas in Fig. 4b and 4c became much wider compared to the previous Fig. 4b and 4c. Please confirm if the statistical test in Fig. 4b and 4c was conducted correctly.?

Ans) There was an error in the statistical significance test in the revised manuscript, so we reconducted the statistical tests for Figure 4b and c (Fig. A). The number of bootstrap runs was increased from 1,000 to 10,000, and the area of statistical significance is nearly identical to the results we firstly submitted.

Fig. A | The changes in ENSO feedback. **a** The regression coefficients of zonal current onto averaged wind stress over the equatorial Pacific (120°E–90°W, 5°S–5°N) during DJF for the PD period. As in **a** but for the difference from the PD period to **b** restoring and **c** IW_be700. The dashed line represents the climatological thermocline depth. Only significant values at the 95% confidence level using the bootstrap test are shown. **d** The regression coefficients of precipitation (shading) and 850hPa wind (vector) onto Niño3 SSTA during DJF for the PD period. As in **d** but for the difference from the PD period to **e** restoring and **f** IW_be700. Stippled areas are regions that are not significant at the 95% confidence level of the Student’s t-test in the PD period and for the differences of restoring and IW_be700 using the bootstrap test.